# Prognostic Value of the Preoperative Prognostic Nutritional Index in Predicting Survival Outcomes After Curative Surgery for Colorectal Cancer

**DOI:** 10.3390/healthcare13233137

**Published:** 2025-12-02

**Authors:** Ozan Barış Namdaroğlu, Ahmet Cem Esmer, Hilmi Yazici, Savaş Yakan

**Affiliations:** 1General Surgery Department, Izmir Tepecik Training and Research Hospital, University of Health Sciences, Izmir 35340, Turkey; ozannamdaroglu@yahoo.com (O.B.N.); savasyakan@gmail.com (S.Y.); 2Surgical Oncology Department, Antalya City Hospital, Antalya 07100, Turkey; ahmetcemesmer@hotmail.com; 3Training Hospital General Surgery Department, Marmara University Pendik Research, Istanbul 34899, Turkey

**Keywords:** colorectal cancer, prognostic nutritional index, survival, inflammation, nutrition, prognosis

## Abstract

**Highlights:**

**What are the main findings?**
Nutritional optimization before surgery might improve outcomes.More careful selection or tailoring of adjuvant therapy may be needed, possibly supported by enhanced supportive care.

**What are the implications of the main findings?**
Enhanced postoperative monitoring and follow-up may help detect complications early or manage recurrence risk more promptly.Discussions with patients about prognosis and treatment trade-offs could incorporate PNI as part of personalized risk communication.

**Abstract:**

**Background:** The Prognostic Nutritional Index (PNI), calculated from serum albumin and lymphocyte count, indicates nutritional and immunological status. Its prognostic significance in colorectal cancer (CRC) is still being assessed. **Methods:** This retrospective study examined 489 patients who received curative resection for colorectal cancer (CRC). According to ROC analysis, patients were split into two groups: those with low PNI (<47.5) and those with high PNI (≥47.5). We compared the clinicopathological features, postoperative outcomes, and survival rates. Kaplan–Meier and Cox regression models were used to look at overall survival (OS) and disease-free survival (DFS). **Results:** A low PNI was strongly related to older age, having a lower BMI, hemoglobin, albumin, and lymphocyte levels (all *p* < 0.001). The low-PNI group had a higher early hospital mortality (4% vs. 1%, *p* = 0.031). Patients with low PNI had a significantly lower five-year OS and DFS (both *p* < 0.001). In multivariate analysis, low PNI independently predicted poor OS (HR = 0.640, *p* = 0.016) and DFS (HR = 0.570, *p* = 0.037), in addition to pathological stage, age, and perineural invasion. **Conclusions:** Preoperative PNI serves as an independent prognostic marker for survival in CRC. A low PNI demonstrates that a patient has low nutritional and immune reserves, which means they are more likely to have worse early and long-term outcomes. Including PNI in preoperative evaluation may help with personalized treatment plans.

## 1. Introduction

Colorectal cancer (CRC) is the third most prevalent cancer and the second leading cause of cancer-related death worldwide, and it is still a significant burden to global health systems [1,2]. While advances in surgical techniques and systemic therapies have improved long-term outcomes, patient survival remains highly heterogeneous. The TNM staging system is the most important determinant of prognosis and treatment planning [3]. However, it might be insufficient to reflect the variability in patient outcomes, particularly within the same stage. This limitation has encouraged researchers to seek additional, easy biomarkers that could improve prognostic classification and identify high-risk patients who may benefit from more aggressive or personalized treatment approaches.

Systemic inflammatory response and nutritional status are increasingly recognized as critical determinants of cancer survival. A weakened nutritional status and persistent systemic inflammation lead to tumor cachexia, immunosuppression, and reduced tolerance to cancer treatments, resulting in poorer survival [4,5]. Various scoring systems, such as the neutrophil-to-lymphocyte ratio (NLR) and the Glasgow Prognostic Score (GPS), have been developed to measure the host’s immune-nutritional status and have demonstrated prognostic value in various cancers, including colorectal cancer [6,7].

The Prognostic Nutritional Index (PNI), developed by Buzby et al., is a nutritional and immunological status-dependent score calculated from serum albumin concentration and peripheral blood lymphocyte count [8]. In recent years, the PNI has been described as a prognostic tool in various malignancies [9,10]. Several studies have suggested that a low PNI is associated with poorer survival in patients with colorectal cancer [11,12]. However, studies on its specific impact on both overall and disease-free survival have not been evaluated in more detail.

In this study, we aim to investigate the prognostic significance of PNI in patients with colorectal cancer in this retrospective cohort. We aimed to evaluate whether pretreatment PNI could be used as an independent predictor of both overall survival (OS) and disease-free survival (DFS) in a relatively large cohort.

## 2. Methods

### 2.1. Study Design

Data regarding patients who underwent curative surgery for colorectal cancer at the University of Health Sciences, Izmir Tepecik Training and Research Hospital’s General Surgery Department, were analyzed retrospectively. The inclusion criteria are as follows: (1) Patients who underwent curative resection for colorectal cancer, (2) Patients who had complete pre-postoperative data, and (3) Patients who had complete follow-up data. The exclusion criteria were as follows: (1) Patients who were below age 18, (2) Patients who received neoadjuvant therapy, and (3) Patients who had a liver disease, autoimmune disease, or any kind of blood disease that may affect the blood count.

### 2.2. Data Collection

Clinicopathological data, including patient demographics such as age, gender, comorbid diseases, Body Mass Index (BMI), American Society of Anesthesiologists (ASA) score, laboratory markers such as Neutrophil, Lymphocyte, and Platelet Count, Hemoglobin, Albumin, Carcinoembryonic Antigen (CEA), Cancer Antigen 19-9, were recorded. All preoperative laboratory data were obtained 1 week before surgery. Operative and postoperative data such as operation type, operation time, postoperative complications, and synchronous metastasectomies were noted from patients’ electronic records. Postoperative complications were evaluated according to the Clavian-Dindo Classification [13]. Histopathological outcomes were obtained from the hospital’s database. PNI was calculated with the following formula: 10 × Albumin Value (g/dL) + 5 × Lymphocyte Count (mL). A Receiver Operator Characteristics (ROC) Curve analysis was performed to determine a cut-off value for PNI. Patients were examined under two groups according to the cut-off value to investigate the operative and prognostic impact of PNI on CRC patients. Survival follow-up was started from the surgical treatment. Overall Survival (OS) is the length of time from the date of surgery until death from any cause. Disease-Free Survival (DFS) is the length of time after surgery during which the patient remains free from any signs of cancer recurrence or new cancer-related events.

The primary outcome of this study is whether PNI is a predictive marker for the prognosis of CRC.

The secondary outcomes of this study are the effect of PNI on postoperative and histopathological results.

### 2.3. Ethical Approval

The study protocol was approved by the Ethics Committee of the University of Health Sciences, Izmir Tepecik Training and Research Hospital (decision date: 6 August 2025, No: 2025/07-03). All procedures complied with the ethical principles of the Declaration of Helsinki. Due to the retrospective nature of the study, the requirement for informed consent was waived.

### 2.4. Statistical Analysis

SPSS version 28.0 (SPSS Inc., IBM, Chicago, IL, USA) was used to perform statistical analysis. The data were presented as mean ± standard deviation (SD), median, and interquartile range (IQR) if the data were not normally distributed. The proportion or frequency was compared between the two groups using Fisher’s exact test or the χ2 test, and differences in continuous variables were evaluated using the Student’s T-test and the Mann–Whitney U test for non-parametric values. Survival curves were estimated with the Kaplan–Meier method, and overall (OS) and disease-free survivals (DFS) were compared using the log-rank test. Cox regression analysis was performed to obtain independent prognostic markers for CRC.

## 3. Results

A total of 489 patients who underwent curative surgery for CRC were included in the analysis. Patients were divided into two groups: low PNI (<47.5, *n* = 201) and high PNI (≥47.5, *n* = 288).

### 3.1. Patient Characteristics

The median age for the low-PNI group was significantly higher than that of the high-PNI group. (Respectively, median 67 (IQR: 16) vs. 62 (IQR: 13), *p* < 0.001). No significant differences were observed in gender distribution or presence of comorbid diseases between groups. Moreover, the rates of Diabetes, Hypertension, Cardiac diseases, and other comorbidities were similar between the groups. BMI was slightly lower in the low-PNI group (median 26 (IQR: 5) vs. 27 (IQR: 5) kg/m^2^, *p* = 0.005). Hemoglobin, albumin, and lymphocyte levels were notably reduced among patients with low PNI (all *p* < 0.001), whereas neutrophil and platelet counts were comparable between groups (Table 1).

### 3.2. Operative and Pathological Findings

Tumor localization, surgical type, and operative time did not differ significantly between groups. The rates of synchronous metastasectomy and synchronous colon tumors were also similar. However, hospital mortality was higher in the low-PNI group (4% vs. 1%, *p* = 0.031). Median hospital stay and major complication rates (Clavien–Dindo grade ≥ 3) were similar among the two groups (Table 2).

Tumor diameter was significantly larger in the low-PNI group (median 50 mm vs. 42 mm, *p* < 0.001). While pathological stage distribution and lymphovascular or perineural invasion rates were similar between groups, patients with higher PNI were more likely to receive adjuvant chemotherapy (73% vs. 65%, *p* = 0.042). The number of harvested and positive lymph nodes did not differ significantly (Table 2).

### 3.3. Survival Analysis

The five-year OS and DFS of the entire cohort were 76.5% and 71.4%, respectively. Kaplan–Meier survival curves demonstrated that patients with high PNI values had significantly better overall and disease-free survival compared to those with low PNI (Figure 1 and Figure 2). Moreover, we performed a Kaplan–Meier analysis regarding TNM stages. Stage 3 and 4 patients had worse OS and DFS, as expected. Log-Rank test showed statistically significant difference (*p* < 0.001 and *p* < 0.001, respectively) (Figure 3 and Figure 4).

In univariable analysis, older age (HR = 1.018, *p* = 0.023), advanced pathological stage (HR = 1.654, *p* < 0.001), perineural invasion (HR = 0.456, *p* < 0.001), lymphovascular invasion (HR = 0.444, *p* < 0.001), and low PNI (HR = 0.556, *p* < 0.001) were significantly associated with worsened overall survival (OS). Multivariable Cox regression analysis confirmed that age (HR = 1.019, *p* = 0.024), advanced stage (HR = 1.555, *p* = 0.008), perineural invasion (HR = 0.653, *p* = 0.031), and low PNI (HR = 0.640, *p* = 0.016) were independent prognostic factors for OS (Table 3).

For disease-free survival (DFS), tumor localization (colon vs. rectum, HR = 0.440, *p* = 0.006), perineural invasion (HR = 0.526, *p* = 0.030), and low PNI (HR = 0.532, *p* = 0.031) were significant in univariate analysis. Multivariate analysis confirmed that tumor localization (HR = 0.448, *p* = 0.007), perineural invasion (HR = 0.553, *p* = 0.046), and low PNI (HR = 0.570, *p* = 0.037) remained independent predictors of DFS (Table 4).

## 4. Discussion

This study demonstrated that a lower PNI is significantly associated with poorer OS and DFS in patients undergoing curative CRC surgery. Patients with PNI < 47.5 had higher age, and were associated with lower hemoglobin, albumin, and lymphocyte levels; moreover, they had higher postoperative mortality compared with those with PNI ≥ 47.5. Notably, multivariable analyses confirmed that low PNI was an independent predictor of both OS and DFS, together with pathological stage, age, and perineural invasion. These findings underscore the prognostic importance of preoperative PNI as a simple, objective, and easily obtained biomarker reflecting the relation between nutritional and immunological status and CRC patients’ prognosis.

Although it was well discussed in existing literature, the relation between colorectal cancer prognosis and inflammation is not certainly described. The PNI score contains serum Albumin level and lymphocyte count. Serum albumin is recognized to correlate with systemic inflammation. Albumin may assist in stabilizing cellular development and DNA replication, buffering various metabolic alterations, and preserving sex hormone balance to mitigate cancer risk [14]. Recent research indicates that hypoalbuminemia signifies a malnutritional and immunosuppressed state in cancer patients, correlating with heightened disease severity, an elevated risk of progression, and diminished survival rates [15]. Lymphocytes are essential in tumor suppression, as they induce cytotoxic cell death and secrete cytokines that inhibit the growth and metastasis of cancer cells. The presence of lymphocytes within the tumor may indicate a favorable prognosis [16]. Memory T cells in colorectal cancer can alter the tumor matrix or tumor cells within the adaptive immune response, thereby diminishing the metastatic potential of tumor cells. The transport characteristics, density, and long-term anti-tumor capability of T-cells likely play a crucial role in regulating tumor recurrence [17]. Tumor-infiltrating lymphocytes in solid tumors demonstrate oligoclonal proliferation, recognition of tumor antigens, and tumor-specific cytolytic activity in vitro, which are associated with enhanced clinical outcomes, including delayed recurrence and reduced mortality. Moreover, Lymphocytes are vital elements of the adaptive immune system, consistently suppressed in tumors via various pathways. Infiltrating lymphocytes have been identified as a significant component of the antineoplastic cellular immune response [18]. A low peripheral lymphocyte count may signify an insufficient immune response to tumors, foster a conducive microenvironment for recurrence, and imply a poor prognosis [19].

Systematic screening of preoperative nutritional status in patients undergoing colorectal surgery and management of patients at high risk of malnutrition with enteral supplementation, prior to surgery, is a common recommendation of the current Enhanced Recovery After Surgery (ERAS) and European Society for Clinical Nutrition and Metabolism (ESPEN) guidelines [20,21]. The ESPEN guidelines also suggest that oral immunonutrition given for 5–7 days before colorectal surgery may be considered, and the potential for targeted, preferably oral, nutritional support for up to 7–10 days to reduce morbidity is particularly emphasized. Recent reviews also support the view that immunonutrition, when administered at adequate doses and for an adequate duration, reduces infectious complications and the need for intensive care [21,22,23]. On the other hand, serum albumin level is a negative acute phase reactant and should not be used as a nutritional indicator [24]. Preoperative hypoalbuminemia (mostly <3.0–3.5 g/dL) should be interpreted as a prognostic risk marker because it is independently associated with wound complications, reoperation, and early mortality in colorectal cancer [25,26]. Serum albumin level alone is not sufficient to diagnose malnutrition; the GLIM (Global Leadership Initiative on Malnutrition) approach recommends meeting both phenotypic (weight loss, low BMI) and etiological (undernutrition, inflammation) criteria [27]. Considering these current data, an optimization process should be initiated with preoperative support in the presence of low albumin and screening-detected malnutrition, in addition to prognostic indices such as PNI. If possible, an ERAS-based, multidisciplinary approach should be adopted, including immunonutrition for 5–7 days before surgery and targeted nutritional support for 7–10 days, to minimize comorbidities (e.g., anemia) [20,21].

Several recent studies have similarly shown that low PNI is linked to poorer survival and more aggressive disease in colorectal cancer. Peng et al. studied 274 patients with stage III colon cancer who underwent curative resection with adjuvant chemotherapy, finding that a low preoperative PNI (≤49.22) was significantly associated with shorter OS and DFS in stage IIIC disease, in multivariate models [28]. Shibutani et al. (2015) showed that both preoperative and postoperative PNI are prognostic for OS in stage II/III CRC; the combination provided even stronger prognostic discrimination [29]. Although postoperative PNI values were unavailable for this study, Shibutani et al. suggest that dynamic postoperative changes may provide additional prognostic insight. Future prospective studies incorporating perioperative immunonutrition, standardized inflammatory markers, and longitudinal PNI assessments may further refine risk stratification. In the large series by Tae Noh et al., including over 3500 CRC patients undergoing curative resection, preoperative PNI thresholds <50 predicted better and worse OS and DFS, also correlating with postoperative complications and adverse pathological features [30]. Our results are also consistent with data in metastatic CRC. A recent study (253 patients) with metastatic disease reported mean PNI = 46.6, showing significantly longer OS in patients with PNI ≥ 46.6 vs. <46.6 (53.06 vs. 38.80 months; *p* = 0.039) [31]. Although metastatic patients differ in baseline risk, this study similarly emphasizes that immunonutrition, as measured by PNI, has broad prognostic relevance.

Lymphocytes are essential in tumor suppression by inducing cytotoxic cell death and secreting cytokines that inhibit the proliferation and spread of cancer cells. The existence of lymphocytes within the tumor may indicate a positive prognostic outcome [16]. Nonetheless, memory T cells in colorectal cancer can alter the tumor matrix or tumor cells in the adaptive immune response to diminish the spreading potential of tumor cells. The transport properties, density, and sustained anti-tumor efficacy of T-cells may be pivotal in regulating tumor recurrence [32]. Tumor-infiltrating lymphocytes in solid tumors demonstrate oligoclonal proliferation, identification of tumor antigens, and tumor-specific cytolytic activity in vitro, which enhances clinical outcomes, including prolonged recurrence intervals and reduced mortality.

In addition to the PNI, our multivariable analysis confirmed various pathological and patient-specific factors as independent predictors of survival. The well-described importance of advanced pathological stage as an essential factor of poor prognosis emphasizes the important effect of tumor burden and anatomical dimension, as determined by the TNM system, on predicted outcomes [3]. Similarly, the identification of advanced age as an independent risk factor for overall survival reflects the interaction between physiological reserve, an increased comorbidity burden, and potentially less aggressive treatment strategies in elderly individuals. The presence of perineural invasion has been identified as a significant independent predictor of both worse OS and DFS, confirming existing research identifies perineural invasion as an indicator of aggressive tumor biology and a precursor of increased recurrence risk [33]. Its validity in our model indicates the inclusion of prognostic information that exceeds conventional staging. Furthermore, for disease-free survival, tumor localization in the rectum was an independent predictor of poorer outcomes compared to colon cancer, which can be attributed to the complex surgical anatomy of the pelvis, higher rates of positive resection margins, and different patterns of recurrence. The fact that the PNI retained its independent prognostic value in a model that included these robust and well-validated factors significantly bolsters the argument for its utility as a complementary biomarker, reflecting a distinct dimension of patients’ immunonutritional status—that is not captured by anatomical or pathological staging alone.

In the present study, the postoperative length of hospitalization (LOS) was similar between the two groups, with a median stay of 5 days. This relatively short duration likely reflects our institution’s use of enhanced recovery after surgery (ERAS) practices, including early mobilization, multimodal analgesia, and optimized perioperative fluid management. Real-world data confirm that structured ERAS implementation is effective in reducing length of stay after colorectal surgery, as shown in a multicenter study across Ontario hospitals that observed a sustained LOS decrease of ~1.05 days after ERAS adoption [34]. Moreover, in a prospective single-center study, perioperative education with explicit expected discharge dates further reduced LOS to a median of 4 days [35]. Other recent data support a mean ERAS-era LOS of around 6 days in randomized and observational studies globally [36].

## 5. Limitations

This study also has some limitations. First, the retrospective design might lead to selection biases. Second, potential confounding by comorbidities, inflammation (e.g., infection), liver function, and other nutritional factors (such as body composition and sarcopenia) may not be fully controlled. The third limitation is that mild or subclinical forms of liver disease, autoimmune disease, or hematologic disorders may indeed influence serum albumin and lymphocyte levels. And finally, although the number of patients is relatively larger, the results of a single center might limit its generalizability.

## 6. Conclusions

In summary, this study found that preoperative PNI is an independent prognostic factor for OS and DFS in colorectal cancer, also relating to short-term operative risk and postoperative mortality. Low PNI indicates patients with diminished nutritional and immunological reserves, who are less likely to undergo or tolerate adjuvant therapy, and who might experience poorer tumor-related outcomes. PNI may serve as a simple, inexpensive screening tool to identify patients at higher risk. For individuals with low PNI, we recommend early nutritional assessment, potential prehabilitation interventions, and closer postoperative monitoring. Further prospective, multicenter studies might enhance the reliability of PNI, and a combination of new markers might result in patient-oriented treatment planning.

## Figures and Tables

**Figure 1 healthcare-13-03137-f001:**
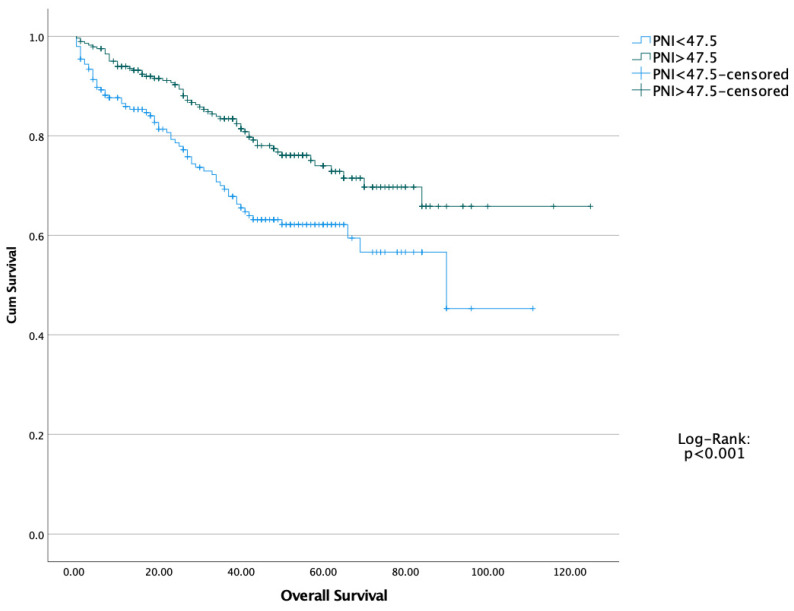
Overall Survival of both Prognostic Nutritional Index (PNI) Groups.

**Figure 2 healthcare-13-03137-f002:**
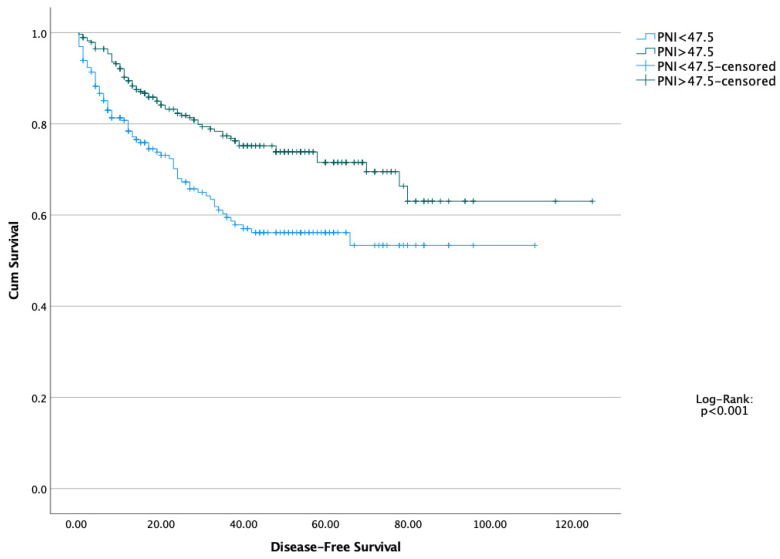
Disease-Free Survival of both Prognostic Nutritional Index (PNI) Groups.

**Figure 3 healthcare-13-03137-f003:**
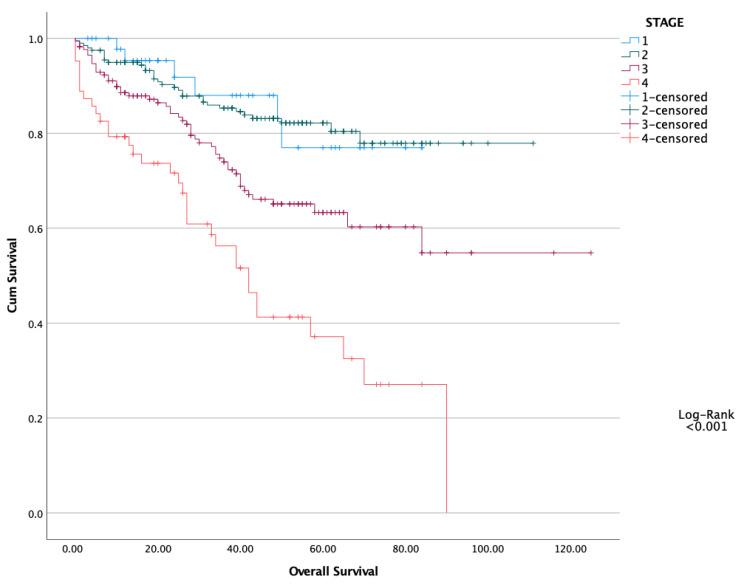
Overall Survival regarding TNM Stages.

**Figure 4 healthcare-13-03137-f004:**
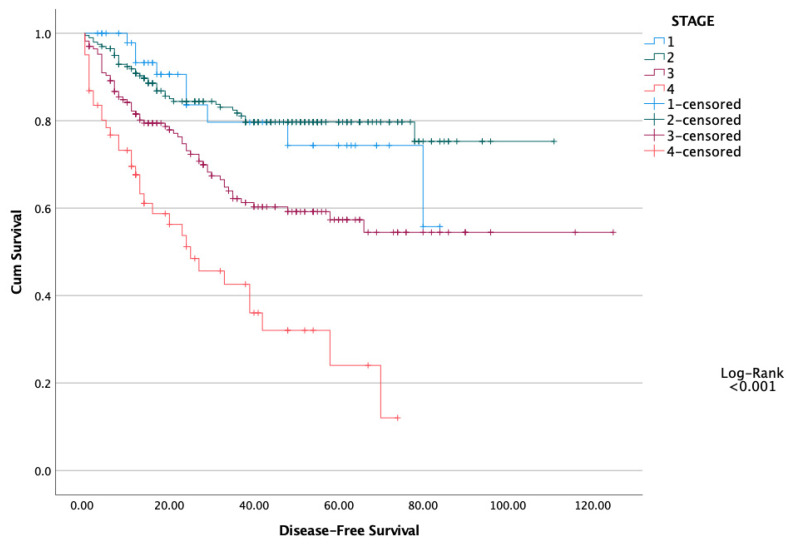
Disease-Free Survival regarding TNM Stages.

**Table 1 healthcare-13-03137-t001:** Basic Characteristics Between Lower and Higher PNI Groups. IQR: Interquartile Range, ASA: American Society of Anesthesiologists.

N: 489	PNI < 47.5 (N: 201)	PNI > 47.5 (N: 288)	*p*
Age [Median (IQR)]	67 (IQR: 16)	62 (IQR: 13)	<0.001
Gender (%)			0.435
Male	108 (54%)	165 (57%)	
Female	93 (46%)	123 (43%)	
Comorbid Diseases			0.411
Presence	124 (62%)	167 (58%)	
Absence	77 (38%)	121 (42%)	
Comorbid Diseases			
Diabetes Mellitus	32 (16%)	56 (19%)	0.318
Hypertension	78 (39%)	94 (33%)	0.159
Coronary Arterial Diseases	23 (11%)	36 (12%)	0.723
Other Diseases	42 (21%)	53 (18%)	0.493
Body Mass Index (kg/m^2^) [Median (IQR)]	26 (5)	27 (5)	0.005
ASA Score			0.108
ASA 1	57 (28%)	105 (36%)	
ASA 2	117 (58%)	156 (54%)	
ASA 3	27 (14%)	27 (10%)	
Neutrophil (mL) [Median (IQR)]	4.6 (IQR: 3)	4.6 (IQR: 2.7)	0.962
Hemoglobin (g/dL) [Median (IQR)]	10.5 (IQR: 2.7)	12.4 (IQR: 2.8)	<0.001
Albumin (g/dL) [Median (IQR)]	3.7 (IQR: 0.5)	4.2 (IQR: 0.4)	<0.001
Lymphocyte (mL) [Median (IQR)]	1.3 (IQR: 0.5)	2 (IQR: 0.9)	<0.001
Platelet (cells ×10^9^ L) [Median (IQR)]	276 (IQR: 126)	269 (IQR: 107)	0.375
Carcinoembryonic Antigen [Median (IQR)]	7 (56)	13 (73)	0.406
Cancer Antigen 19-9 [Median (IQR)]	14 (76)	21 (86)	0.111

**Table 2 healthcare-13-03137-t002:** Operative and Pathological outcomes in both groups. IQR: Interquartile Range, CD: Clavian-Dindo, SD: Standard Deviation, *: Familial Adenomatous Polyposis.

N: 489	PNI < 47.5 (N: 201)	PNI > 47.5 (N: 288)	*p*
Tumor Localization			0.985
Right Colon	63 (31%)	97 (34%)	
Transverse Colon	22 (11%)	31 (11%)	
Left Colon	55 (27%)	78 (27%)	
Rectum	54 (27%)	72 (25%)	
FAP */Synchrone	7 (4%)	10 (3%)	
Operation			0.552
Right Hemicolectomy	78 (39%)	121 (42%)	
Left Hemicolectomy	22 (11%)	23 (8%)	
Anterior Resection	36 (18%)	45 (16%)	
Low Anterior Resection	41 (20%)	80 (28%)	
Abdominoperineal Resection	16 (8%)	7 (2%)	
Total Colectomy	8 (4%)	12 (4%)	
Operation Time (minute) [Median (IQR)]	120 (60)	120 (60)	0.993
Syncrone Metastasectomy	23 (11%)	32 (11%)	0.494
Syncrone Colon Tumor	6 (3%)	8 (3%)	0.892
Complications (CD ≥ Grade 3)	17 (8%)	17 (6%)	0.342
Hospital Stay (Days) [Median (IQR)]	5 (2)	5 (2)	0.800
Hospital Mortality	8 (4%)	3 (1%)	0.031
Pathological Stage			0.418
Stage I	16 (8%)	36 (12%)	
Stage II	83 (41%)	119 (41%)	
Stage III	74 (37%)	98 (34%)	
Stage IV	28 (14%)	35 (12%)	
Perineural Invasion	70 (35%)	87 (30%)	0.282
Lymphovascular Invasion	131 (65%)	193 (67%)	0.672
Tumor Diameter (mm) [Median (IQR)]	50 (30)	42 (30)	<0.001
Adjuvant Chemotherapy	130 (65%)	214 (73%)	0.042
Adjuvant Radiotherapy	24 (12%)	35 (12%)	0.487
Harvested Lymph Nodes [Median (IQR)]	19 (13)	17 (10)	0.459
Tumor Positive Lympph Nodes (mean ± SD)	1.6 (±2.7)	1.4 (±2.9)	0.904

**Table 3 healthcare-13-03137-t003:** Univariable and Multivariable Overall Survival Analysis for Colorectal Cancer. ASA: American Society of Anesthesiologists, PNI: Prognostic Nutritional Index. Significant Values are written in bold.

N: 489	HR	95% CI	*p*	HR	95% CI	*p*
Gender	0.821	0.572–1.179	0.286			
Age	1.018	1.002–1.034	**0.023**	1.019	1.003–1.036	**0.024**
Comorbid Diseases	1.326	1.125–2.625	0.125			
Hemoglobin (g/dL)	0.995	0.920–1.077	0.906			
Tumor Localization (Colon vs. Rectum)	1.114	0.724–1.716	0.623			
ASA Score	0.638	0.383–1.063	0.084			
Stage (1–2 vs. 3–4)	1.654	1.366–2.003	**<0.001**	1.555	1.120–2.157	**0.008**
Tumor Positive Lymph Nodes (N0 vs. N+)	0.434	0.302–0.623	**<0.001**	1.082	0.585–2001	0.803
Tumor Diameter	1.006	0.999–1014	0.101			
Perineural Invasion	0.456	0.320–0.650	**<0.001**	0.653	0.443–0.962	**0.031**
Lymphovascular Invasion	0.444	0.286–0.689	**<0.001**	0.701	0.429–1.145	0.156
Adjuvant Chemotherapy	1.040	0.697–1.550	0.848			
Adjuvant Radiotherapy	0.959	0.538–1.708	0.886			
PNI (<47.5)	0.556	0.390–0.792	**<0.001**	0.640	0.445–0.922	**0.016**

**Table 4 healthcare-13-03137-t004:** Univariable and Multivariable Disease-Free Survival Analysis for Colorectal Cancer. ASA: American Society of Anesthesiologists, PNI: Prognostic Nutritional Index. Significant Values are written in bold.

N: 489	HR	95% CI	*p*	HR	95% CI	*p*
Gender	0.754	0.418–1.1359	0.348			
Age	0.991	0.968–1.015	0.466			
Comorbid Diseases	1.3216	1.025–1.912	0.332			
Hemoglobin (g/dL)	0.999	0.878–1.137	0.988			
Tumor Localization (Colon vs. Rectum)	0.440	0.245–0.791	**0.006**	0.448	0.249–0.806	**0.007**
ASA Score	1.773	0.519–6.058	0.361			
Stage (1–2 vs. 3–4)	1.309	0.981–1.746	0.067			
Tumor Positive Lymph Nodes (N0 vs. N+)	0.640	0.361–1.135	0.127			
Tumor Diameter	1.005	0.993–1017	0.383			
Perineural Invasion	0.526	0.295–0.939	**0.030**	0.553	0.309–0.989	**0.046**
Lymphovascular Invasion	1.404	0.790–1.404	0.248			
Adjuvant Chemotherapy	1.051	0.555–1.993	0.878			
Adjuvant Radiotherapy	1.187	0.425–3.318	0.744			
PNI (<47.5)	0.532	0.300–0.944	**0.031**	0.570	0.320–0.916	**0.037**

## Data Availability

The data presented in this study are available on request from the corresponding author due to ethical and legal concerns of data privacy.

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
