# Peer review of "Prognostic Value of the Preoperative Prognostic Nutritional Index in Predicting Survival Outcomes After Curative Surgery for Colorectal Cancer"

_healthcare, 2025, doi:10.3390/healthcare13233137_

Round 1

Reviewer 1 Report

Comments and Suggestions for Authors

The study was designed as a retrospective observational analysis and conducted at a single center (the hospital name is not explicitly stated). A total of 489 patients were included in the analysis. Inclusion criteria comprised undergoing curative resection, availability of complete pre- and postoperative data, and complete follow-up. Patients under the age of 18, those who received neoadjuvant therapy, and individuals with liver, autoimmune, or hematologic diseases were excluded. Although the sample size appears reasonable, the absence of a power analysis and the lack of specification regarding the patient selection period are notable limitations.

The statistical methods were appropriately chosen. The calculation of the Prognostic Nutritional Index (PNI) follows the standard formula in the literature: PNI = 10 × (albumin, g/dL) + 5 × (lymphocyte count, /mL). The PNI cutoff value was determined to be 47.5 based on ROC curve analysis, and patients were categorized into two groups accordingly. Kaplan–Meier curves and the log-rank test were used for survival analysis, while multivariate analysis was performed using Cox regression.

The presented results are consistent with the existing literature. Previous studies have similarly reported that low preoperative PNI is associated with poorer survival in colorectal cancer patients. The current study supports these findings. The authors have referenced up-to-date literature, including global carcinoma statistics and the latest ERAS/ESPEN guidelines (2024–2025), which strengthens the study's relevance.

The study's novelty lies primarily in its relatively large sample size and its concurrent evaluation of both overall survival and disease-free survival, along with its emphasis on the relationship between PNI and early postoperative mortality. Nonetheless, the core findings are largely aligned with those of prior similar studies. Therefore, the scientific contribution of the article can be interpreted as confirmatory of existing evidence and as providing data specific to the Turkish patient population. Overall, the manuscript aligns with current knowledge and highlights the potential of PNI to enhance prognostic stratification in colorectal cancer.

In terms of structure, the article largely follows a scientific style and is formatted according to MDPI standards. Sections such as the title, abstract, introduction, methods, and results are presented in a logical and clear order. However, there are several issues with the quality of writing.

In 66. and 172. lines repeated abbreviation of “PNI” should be removed. In line 66 “Buzby at all.” should be corrected.

In line 68 71, used parentheses rather than square brackets for numbered references.

In line 79-80 “Hospital’s General Surgery Department” is not specified. Authors should clarify the data source and if it is multicentered study.

Line 176 “prevalant” should be corrected.

The authors should specify the starting time point for overall survival more clearly. It can begin with diagnosis or initial treatment.

Discussion emphasizes how albumin and lymphocyte may affect prognosis via their essential protective functions. However, exclusion criteria involve pathologies that alter immune and blood system. In this perspective, all other factors that perturb these systems also may affect the overall survival rate. Suggestion is, changing the perspective or exclusion criteria. Additionally, neoadjuvant therapies target to enhance immune response which is an exact point of allegation of PNI score related survival. After all exclusion criterions may mask the real factorial results. If the starting point of the overall survival rate is before resection, it could be discussed as more specific to cancer-related immune system failure or exhaustion.

The manuscript is noteworthy for demonstrating the prognostic value of the PNI as a preoperative indicator of nutritional and immune status in colorectal cancer. Its strengths include a reasonably sized patient cohort, references to up-to-date literature, and the application of standard statistical methods. The finding that PNI independently affects both overall survival and disease-free survival is of clinical significance. However, the study's limitations include its retrospective, single-center design, the lack of specification regarding the patient selection period, and the omission of certain potential confounders (such as receipt of adjuvant therapy) from the analysis. Additionally, repetitive statements in the manuscript and the presentation of some statistical outcomes-such as the hazard ratio for disease-free survival with a 95% confidence interval of 0.320–1.013-require clarification.

In conclusion, the study is of acceptable quality but requires revisions before it can be submitted.

Reviewer 2 Report

Comments and Suggestions for Authors

Congratulations to the authors for the chosen topic, which is current and has sparked numerous debates in the literature.

The article is well structured and argued, but important changes are needed to increase its scientific value.

There are data in the results chapter that create confusion in their interpretation in the discussion chapter.

From my point of view, the authors should supplement the results chapter with information about comorbidities - these data must correlate with the ASA score and can explain at the same time whether they can correlate with the average postoperative survival rate.

Also in the results chapter, it is not clear whether the data are number of patients/percentage/median. These data need to be clarified.

I believe that it would also be useful to correlate the TNM stage with OS and DFS.

The length of hospitalization is identical between the two groups, but the values ​​are not understood - are there only 5 days of postoperative hospitalization? And the length of hospitalization can be discussed and correlated in the general context of the discussions.

It would be useful to add other parameters, especially for the evaluation of postoperative survival: renal function, liver function, ions, etc.

The references chapter can be improved with other recent articles.

Reviewer 3 Report

Comments and Suggestions for Authors

1. Research Methodology and Participant Selection:

 The study is retrospective and conducted at a single center. The authors recognize this as a constraint. What particular measures were implemented to reduce selection bias in the patient cohort? Were there any modifications in surgical techniques, adjuvant therapy protocols, or nutritional support practices over the research period that could obscure the results?
 Patients with liver illness, autoimmune disease, or hematological problems are excluded from the criteria. What methods were employed to diagnose and validate these conditions? Can less severe manifestations of these illnesses, which do not fulfill the stringent diagnostic criteria, nonetheless affect the PNI and outcomes?

 The study specifies "comprehensive pre- and postoperative data" as a criterion for inclusion. What specific data points were necessary for inclusion, and what percentage of potentially eligible patients were removed owing to insufficient data? Could this exclusion engender bias?
 2. PNI Threshold Value:

 The PNI cut-off value of 47.5 was established by ROC analysis. Was this cut-off figure predetermined, or was it established subsequent to data analysis? If so, how was overfitting mitigated?
 To what extent does the conclusion depend on the selected cut-off value? Did the authors conduct a sensitivity analysis by evaluating alternative plausible cut-off values?
 The selected cut-off is somewhat low in relation to other studies referenced in the debate. Can this discrepancy be attributed to differences in patient demographics, laboratory techniques, or other variables?
 3. Statistical Examination:

 The authors employed univariate and multivariate Cox regression analyses. Were the assumptions of Cox regression, such as proportional hazards, sufficiently evaluated?  What alternate solutions were contemplated if not achieved?
 In the multivariate analysis, various factors (age, stage, perineural invasion, and PNI) emerged as independent predictors of overall survival (OS). Is there a possibility of multicollinearity among these variables? What methods were employed for assessment and resolution?
 Table 4 indicates that in the multivariate analysis, the PNI exhibits a p-value of 0.037 and a hazard ratio of 0.570, with a confidence interval of 0.320-1.013, which includes 1 and is marginally significant at the 0.05 level. Is the study sufficiently powered to identify the impact of PNI on DFS?
 4. Discourse and Analysis:

 The discourse emphasizes the importance of immunonutrition and Enhanced Recovery After Surgery (ERAS) protocols.  Nevertheless, the study appears to lack detailed consideration of the particular nutritional therapy administered to patients.  In what ways could variations in nutritional support have affected the outcomes?
 The authors recognize that any confounding due to comorbidities and inflammation may not be entirely mitigated. What precise data about comorbidities and inflammatory markers were gathered, and how were they integrated into the study, if at all?
 The research examines preoperative PNI.  Would the integration of postoperative PNI readings yield supplementary predictive insights, as proposed by Shibutani et al. (2015)? What were the reasons for or against this consideration?
 5. General Considerations:

 The manuscript needs copyediting for clarity and grammatical accuracy. There are variations in the presentation of p-values and confidence intervals.
 The clinical implications of the findings could be enhanced. In what manner should clinicians include PNI in their routine practice? What particular therapies should be contemplated for patients with a poor Prognostic Nutritional Index (PNI)?

Round 2

Reviewer 1 Report

Comments and Suggestions for Authors

Authors amended all the suggestions. The manuscript can be accepted in this form.

Reviewer 2 Report

Comments and Suggestions for Authors

Congratulations to the authors for the effort made to improve the scientific quality of the article.

I am glad that my suggestions were useful and accepted by the authors to bring significant improvements to the article.

From my point of view, the article can be published.

Reviewer 3 Report

Comments and Suggestions for Authors

The comments were ok.